# Depression among the Non-Native International Undergraduate Students Studying Dentistry in Bangladesh

**DOI:** 10.3390/ijerph18115802

**Published:** 2021-05-28

**Authors:** Russell Kabir, Samia Naz Isha, Mohammad Tawfique Hossain Chowdhury, Nazeeba Siddika, Shah Saif Jahan, Arup Kumar Saha, Sujan Kanti Nath, Mohammed Shahed Jahan, Madhini Sivasubramanian, Ilias Mahmud, Ehsanul Hoque Apu

**Affiliations:** 1School of Allied Health, Faculty of Health, Education, Medicine and Social Care, Anglia Ruskin University, Chelmsford Essex CM1 1SQ, UK; ssj151@student.aru.ac.uk; 2CAPABLE-A Cambridge-Led Programme in Bangladesh, University of Cambridge, Cambridge CB2 1TN, UK; sni25@medschl.cam.ac.uk; 3Department of Dental Public Health, Sapporo Dental College, Dhaka 1230, Bangladesh; tawfique@sdch.edu.bd (M.T.H.C.); knsujan@yahoo.com (S.K.N.); 4Center for Environmental and Respiratory Health Research (CERH), Faculty of Medicine, University of Oulu, 90014 Oulu, Finland; nazeeba.siddika@oulu.fi; 5Department of Dental Public Health, City Dental College, Dhaka 1229, Bangladesh; arupcdc@yahoo.com; 6Department of Dental Public Health, Update Dental College, Dhaka 1711, Bangladesh; shahed.jahan5@gmail.com; 7Programme Manager (MSc Nursing and PH), The University of Sunderland in London, London E14 9SG, UK; madhini.sivasubramanian@sunderland.ac.uk; 8Department of Public Health, College of Public Health and Health Informatics, Qassim University, Al Bukairiyah 52741, Saudi Arabia; i.emdadulhaque@qu.edu.sa; 9Department of Biomedical Engineering, Institute of Quantitative Health Science (IQ), Michigan State University, East Lansing, MI 48824, USA; hoqueapu@msu.edu

**Keywords:** international student, depression, Bangladesh, dental, CES-D, undergraduate

## Abstract

Background: Bangladesh has been attracting international students with interests in various subjects recently. Every year students from different parts of the world come to study undergraduate and postgraduate courses, mostly at private universities in Bangladesh. This study evaluates the depression status among international students who are studying dentistry in Bangladesh. Methods: This cross-sectional survey was conducted among International undergraduate dental students who enrolled in the Bachelor of Dental Surgery program in nine public and private dental colleges in Bangladesh. Participants were selected using a convenience sampling method. A total of 206 students completed the survey where 78.5% of them were female students and 21.5% students were male, and a CES-D 10-item Likert scale questionnaire was used for data collection. The Cronbach alpha for the 10-item CES-D scale for this population is 0.812. Results: The majority of the students (79.5%) are below 24 years of age with a mean age of 23.22 years and standard deviation of 2.3, and are students who cannot communicate well in Bengali (Bangla), about 60% of them have experienced depression. About 77.3% (*p* < 0.00) of the international students having financial difficulties exhibited depression. The international students who went through financial problems were two times more likely to suffer from depression (OR = 2.38; *p*-value < 0.01). Conclusion: This study tried to highlight the struggles faced by international students in Bangladesh studying dentistry. It is evident from the findings that several factors influence students’ mental well-being during demanding dental education years.

## 1. Introduction

International undergraduate students (IUSs) are defined as those students that have crossed borders to obtain education in a non-native country, away from home, family, and loved ones [1]. Recently, Bangladesh has been attracting students from different parts of the world for various undergraduate programs [2]. The inflow of IUSs is beneficial for both the host country and the students; IUSs benefit the host country by contributing “intellectual capital,” whereas the IUSs receive education and a degree. Nonetheless, IUSs face innumerable challenges when moving to a completely unknown non-native country [3]. Beyond the difficulties related to the educational institution, adjusting to a new place may present socialization stress among the students [3,4]. The IUSs encounter events and stressors beyond that of native students that have an impact on their mental health [4]. They have to deal with these problems without their family or familial support. The inability to cope with lectures and classroom activities and the most significant language barrier may result in anxiety or depression among the students [3].

Mental health is a growing concern for low and middle-income countries, and it is a tremendous public health concern worldwide. According to the World Health Organization (WHO), depression is defined as “a common mental disorder, characterized by sadness, loss of interest or pleasure, feelings of guilt or low self-worth, disturbed sleep or appetite and feelings of tiredness and poor concentration. It can be long-lasting or recurrent, substantially impairing a person’s ability to function at work, school or cope with daily life” [3,5,6,7]. Mental conditions may reduce an individual’s lifespan by 12.3% adjusted for disabilities, and 31% of all years of life at all ages. The new updated WHO figures state that 300 million people are now dealing with depression and that between 2005 and 2015, this rose by 18% [8]. A study from Bangladesh revealed depression prevalence of 52.5 to 54.3% among students enrolled in medical schools [5].

Globally, medical and dental schooling is perceived as a highly stressful education, where students face a higher level of stress, anxiety, and depression compared to peers studying other subjects [9,10,11]. In particular, research shows that dental students are prone to higher burnouts compared to medical students as it involves patient interaction during both theoretical and clinical course elements [12]. Obtaining an undergraduate academic degree in dentistry requires tedious education and demands to acquire deep knowledge on the field. The course usually runs for five years, with the last two years for clinical training, and one year of internship follows upon graduation. An undergraduate dental student must acquire theoretical knowledge and practical experience and develop interpersonal skills, which are judged at the end of each academic year through oral, written, and practical examinations. These persistent academic workloads and non-academic stressors, e.g., whole working day, coordinating with faculties, and administrative formalities, are generally stressful for students [6,13,14,15]. With this in mind, the extensive periods of self-study, minimal sleep, and hours of professional education may, in turn, contribute to social deprivation, psychological exhaustion, and depression. This burden may have a detrimental effect on their work and personal life. Early-stage identification of depression is vital for minimizing suicide deaths and intentional self-harm.

Besides the regular academic and non-academic workload that every student must follow, overseas students may face various other difficulties. For instance, language and cultural differences may cause a considerable barrier. They may also endure financial crisis, unclear immigration status, homesickness, and alleged inequality above the usual university issues [16]. Students, especially those studying in foreign countries, who fail to adapt to changes and their new surroundings are more vulnerable to depression [17,18]. Several studies have shown that depression is extremely prevalent among international students [17,19,20]. Previous studies on IUSs have been conducted primarily in the developed countries. Several studies have recently emerged on the topic related to depression among students studying in government-funded public versus self-funded private universities [17,21]. It is also crucial to understand the extent of the effect on mental health as a result of studying dentistry abroad with different cultures, food habits, and adaptation among IUSs. The number of international students studying in Bangladesh has increased since 2012 and contributed to the local educational system but there is a research gap regarding their experience and learning outcome [22]. However, no such study has been conducted worldwide among non-native international undergraduate students studying dentistry and its effects on mental health. Hence, for the first time, this study evaluates the depression status among international students studying dentistry in Bangladesh through a cross-sectional study design. Here, we hypothesized that a change in the sociocultural environment, financial security, and structure of course curriculum has an impact on the mental health of non-native IUSs studying dentistry in Bangladesh.

## 2. Methods

In this original study, we have utilized a web-based cross-sectional survey design. This study’s target group was IUSs who came to study for a Bachelor of Dental Surgery qualification in public and private dental colleges in Bangladesh. The criteria were used to recruit the respondents—(i) international undergraduate dental students (IUDSs), (ii) enrolled in the Bachelor of Dental Surgery (BDS) program after completion, which leads towards graduation in dentistry. No exclusion criteria were used. Data were collected from October to December 2020 from the following selected dental colleges—Dhaka Dental College and Hospital, City Dental College, Mandy Dental College and Hospital, Pioneer Dental College and Hospital, Rangpur Dental College, Update Dental College and Hospital, University Dental College and Hospital, MH Samorita Hospital and Medical College Dental Unit, and Sapporo Dental College and Hospital. The researchers used a convenience sampling method to recruit the students by using their academic and professional network to find a gatekeeper from each of the dental colleges to send the Google form link. The researchers used a snowballing sampling process and requested the IUDSs to circulate the Google form weblink with other IUDSs in their respective colleges. A total of 206 students from the previously mentioned nine dental colleges submitted their responses. Among them, five students did not consent to participate in the survey, and one of them did not thoroughly fill out the questionnaire. Hence, the final sample size of the survey was 200 IUDSs.

The survey questionnaire was adapted from similar research conducted in Bangladesh [4]. The online questionnaire had three parts—background information, challenges faced, and depression measurement. The background information included—age, gender, year of study, marital status, religion, living arrangements in Bangladesh, communication in Bengali (Bangla), and cooperation from the teacher or faculty member. The various challenges faced included problems adapting to local Bangladeshi food, homesickness, health problems, financial difficulty, living condition, and transport facility.

For the assessment of depression, the Centre for Epidemiological Studies Depression Scale (CES-D-10) 10-item Likert scale questionnaire was used [23]. The questionnaire contains three items on depressed affect, five on somatic symptoms, and two on positive affect. Each item has the same numerical rating scale, and 0 represents “rarely,” 1 means “some,” 2 represents “occasionally,” and 3 illustrates “most.” The scoring was reversed for items 5 and 8, which are positive affect statements [24]. Total scores can range from 0 to 30 and the cut-off scores for depressive symptoms were 10 or higher. The Cronbach alpha for the 10-item CES-D scale for this population was 0.812.

We estimated the prevalence of depression. Descriptive analyses were performed to present the prevalence of depression by background characteristics. We investigated the association between depression and background characteristics of the international students with the Chi-squared tests. While the association between the challenges faced by the international students and depression was investigated by multivariable logistic regression analysis. We reported the odds ratio (OR) with a 95% confidence interval (CI) for the multivariable logistic regression analysis. For both the Chi-squared tests and logistic regression analysis a *p*-value of <0.05 was considered statistically significant.

The study was reviewed and approved (SDC/C-7/2020/764) by the Research Ethics Committee of Sapporo Dental College and Hospital, Dhaka, Bangladesh. Participation in this survey was voluntary and informed consent was sought before the study. The objectives of the research were clearly explained and the anonymity and confidentiality of the information were assured. No incentives were offered to the participants. Data analysis was performed using IBM SPSS version 26 (SPSS Inc., Chicago, IL, USA).

## 3. Results

The students’ ages ranged from 18 to 32 years, with a mean age of 23.22 ± 2.31. The majority of the students (79.5%) were below 24 years of age as presented in Table 1. About 52.2% of female students shared that they were experiencing depression, and approximately 53.1% of students from religious groups other than Muslim have experienced depression. Third (3rd) year dental students (61.1%) had experienced depression compared to the students from different academic years. For students that cannot communicate well in Bengali (Bangla), about 60% of them experienced depression. Of the students that expressed problems regarding their studies, almost 63% of them revealed that they suffered from depression. Poor cooperation is also linked with students suffering from depression. Of the students that had poor academic interactions with their teachers, about 69.2% complained about suffering from depression. The students living with friends said they did not suffer from depression compared to the students living alone (75%), and students living with others (78.6%) have expressed their concerns with depression.

The vast majority of the international students (70.2%) had problems adapting to Bangladeshi food also experienced depression. About 80.5% of students suffered from depression due to homesickness. The vast majority of the students who had health problems (65.5%), financial difficulty (77.3%), and problems with the living condition (79.7%) complained about depression. Of the students who found it challenging to use the transport facility in Bangladesh about 65.5% of them suffered from depression. Homesickness, health problems, financial difficulty, and living conditions were significantly associated with depression as shown in Table 2. Also, the Cramer’s V coefficient suggests that homesickness and health problems have a moderate association with depression, and problems adapting to local Bangladeshi food, financial difficulty, accommodation, and transport problems have a substantial relationship with depression.

The multivariable logistic regression analysis findings are shown in Table 3. The logistic regression analysis revealed that the international students who had difficulties communicating in the local language were 2.28 (95% CI: 1.13–4.61) times more likely to experience depression than those without such difficulties. The odds of depression among the homesick international student were 2.75 (95% CI: 1.38–5.47) times greater than the odds of depression among the students who did not feel homesick during their stay in Bangladesh. International students that experienced any health problems had 3.57 (95% CI: 1.86–6.87) times greater odds of experiencing depression compared to the students without any health problems. The international students who went through financial difficulties were 2.62 (95% CI: 1.31–5.27) times more likely to experience depression than those who did not face any financial problems. Nagelkerke’s R^2^ 0.307 indicates that 30.7% of the variance in the outcome variable is explained by the independent variables included in this multivariable logistic regression model. The Hosmer and Lemeshow test (*p* = 0.783) indicated a good model fit.

## 4. Discussion

Our study using the CES-D scale reveals that 51.2% of the international students, studying dentistry in Bangladesh, suffer from depression. The prevalence of depression increases with advancing years in education. In Bangladesh, undergraduate dentistry’s new curriculum [25] consists of five years, with the first two and half years dedicated to knowledge-based education followed by practical and clinical teaching. After successful completion of the BDS course, they must complete a rotatory-based internship for one year. The increase in distress can be due to the transition from pre-clinical to clinical stages, as demonstrated in other studies [6,26]. Most studies reported that international female students studying medicine or dentistry are more likely to suffer from anxiety and depression. The difference among female and male students varies from 53.9% to 30% and 46.1% to 16%, respectively [27,28]. However, in our study, there was not any vast difference among the gender regarding depression among the students, with 52.2% female compared to 48.8% male students.

In our study, 77.3% (*p* < 0.00) of the international students having financial difficulties exhibited depression. This is consistent with several studies conducted on similar populations that found financial responsibilities or status to be significantly related to depressive moods [27,29,30]. Although it was also seen when students are financially secured with scholarships and funding, they tend to be less susceptible to depression than those with none [17] as private universities or institutions are more expensive than public or government-funded ones [16]. The tuition fee for international students is more than that for national students; this helps generate additional revenue for the institution [1]. The higher tuition fees and the added expense of living puts the students under tremendous stress [19].

A significant change in a student’s life is the transition from living with family to living away from their social network of family, friends, and neighbors in a non-native foreign country [3,16,18]. Adapting to a new environment and the absence of familiar surroundings tend to affect students’ mood and stress levels [31]. As stated in two studies by Moeini, Biasi, and their colleagues, “depression is one of the homesickness components,” as is reflected in our study [32,33]. Those who are homesick are 2.5 times more likely to be depressed. The state of accommodation plays an important role apart from homesickness on an international student’s mental health. Not getting the desired place of living on time or the unavailability of a dormitory act as stressors on the newly arrived students [3]. Our study found that a higher proportion of those facing accommodation problems experience depression, however, this result was not statistically significant.

Stressors such as the authoritarian style teaching/education and poor feedback from faculties might cause great distress among students [3,33]. In Asian educational institutes, the relation between teacher and student is very formal, which may hinder better support and cooperation. A study conducted in a Greek dental school found that an informal academic connection between student and teacher provides a positive environment [33]. Our research shows that students who received full intellectual interaction did not suffer from depression in contrast to those with poor academic interaction. In most studies, one of the most critical hurdles for a non-native international students studying healthcare is the inability to communicate [18,34]; the adaptation to the host country’s language. This affects the social interaction with other students and generates a communication gap between the institution and hostel (accommodation) administration who speaks only the host native language [3]. This language barrier affects a student’s ability to understand class lectures and assignments, the confidence to participate and communicate in class, and may impact performance during oral examinations. A similar study conducted in China found that international students are not appropriately assigned due to patient care, history taking, and conveying treatment plans for communication insufficiencies [35]. Even among international students studying in Australia and in the USA, which are officially English speaking countries, studies have reported differences in language proficiency and multilingualism concerning their academic experience and results [36,37,38].

As dentistry is a clinical and research-oriented subject, the post-graduate career development issues may also cause depression among the IUDSs graduating from dental colleges in Bangladesh. Whether the graduates will continue to further their academic or research career paths in Bangladesh or return to their native countries may seem challenging to solve in a minimal amount of time after graduation. As a non-native dentist compared to their fellow native colleagues, the fresh international graduate dentists in Bangladesh may face challenges and have difficulty in learning clinical skills and gaining trust among the local patients during their mandatory internship training periods. In an exciting study, Cohen and colleagues [37] explored the clinical preparedness range, communication ability, social comforts, and clinical confidence between USA-born and non-USA-born clinical radiation sciences students, and the findings were intriguing. Although the academic and research environment is more structured for international students in the USA than in Bangladesh and the widely used communication language between the students, clinicians, and patients in English, the student’s place of birth was an influencing factor in each clinical preparedness domain [37].

We must consider that international students undertake different English proficiency exams before continuing their education in the USA. Even they felt less prepared clinically than their native USA-born colleagues. In Bangladesh, an IUD must communicate both in Bengali and English to complete such a challenging clinical degree in dentistry. Furthermore, even if an IUD can learn Bengali, it takes time to get used to the typical local dialect, which varies according to localities and races. They can communicate in social and academic life, even with their patients, during clinical learning sessions. The patients may only explain the signs in their native and local Bengali dialects. A study revealed that foreign-trained dentists interested in seeking employment in the USA face numerous challenges, including complex admission processes, high tuition costs, immigration barriers, and cultural differences [21]. Even a study conducted on non-native international students pursuing post-graduate education in Finland exhibited that international students face academic, social, financial, and mental health challenges and have concerns about their future career in a country that is officially a non-native-English-speaking one [18]. Similarly, in Bangladesh, an IUDS faces similar challenges and becomes worried, and may even get depressed regarding their potential dentistry career. After graduation, they can be a massive asset to Bangladesh’s local healthcare sector and contribute further as trained international clinicians and innovative researchers. As Bangladesh moves towards a developing country and its economic growth attracts more global investments, it will encourage foreign employees to immigrate for their professional career development. The international dentists trained in Bangladesh can support and serve the international community in the coming future. The government and health care authorities should make policies and guidelines for a smoother career path for the IUDSs who will graduate from dental colleges in Bangladesh. Increasing the diversity in the dental profession in Bangladesh to match the native population might improve dentistry access for the locals, international residents, and minorities, hence reducing dental care disparities.

## 5. Strengths and Limitations

This study was conducted in Bangladesh where English is not an official language and is seldom spoken during day-to-day conversations in the community. Previous research on non-native international students’ well-being shows that even in officially native-English-speaking countries, one-to-one language exchange was identified as a significant barrier for communication for students [34,36,37]. We have reported that the non-native IUDSs face a language barrier that is even greater while studying a clinically oriented subject such as dentistry in Bangladesh. The data support previously published findings on non-native international students enrolled in universities from non-native-English-speaking countries such as Finland and China [18,35]. Although, their educational infrastructure and pattern are traditionally more established for hosting non-native international students. Additionally, primary data of the current study was collected from IUSs from multiple public and private dental universities of Bangladesh. The findings provide a broad view of student’s perspectives and without any doubt, they will provide the necessary feedback for the educational institutions, foreign ministry, and respective international diplomatic representatives. To date, this will be the first study conducted on non-native international undergraduate dental students (IUDSs), who will be engaged in a very skilled and technically advanced profession dealing with patients, both in Bangladesh, their home country, or even in a third country where they may migrate for higher studies and professional development.

The depressive symptoms were self-reported and measured through a survey using the validated Centre for Epidemiological Studies Depression Scale (CES-D-10), which may be considered as a limitation of the study. A clinical interview of the IUSs might have given a clearer impression of the actual mental health status. Also, a study on faculty members would give a better overview regarding the IUDSs’ academic performance and possible ways to improve their experience while studying in Bangladesh. This study was conducted in a very difficult global crisis, during the ongoing lengthy coronavirus disease 2019 (COVID-19) pandemic and partial lockdown situation, so collecting more detailed information about the specific issues highlighted in the study was not practically possible. We conducted studies on native Bangladeshi dentists’ knowledge on COVID-19 [39] and clinical practice during the pandemic, and on intentions to receive the COVID-19 vaccine of the general Bangladeshi population [40]. However, we could not collect these data for the non-native IUDSs due to time limitations and the closure of institutes due to lockdown. In both cases, this vital information collected from the IUDS participants would have strengthened the study findings and hypothesis.

## 6. Conclusions

This study tried to highlight the struggles faced by international students in Bangladesh studying dentistry. It is evident from the findings that several factors influence students’ mental well-being during demanding dental education years. The educational institute must support housing, visa issues, and management of challenges faced due to cultural differences for the international students. Helping students to integrate into culture also requires attention from teachers and fellow students. Class lectures should be conducted in a medium that allows each student to understand the content. Educational institutes should offer or guide IUSs to take up a language course for beginners. Also, Bangladeshi diplomatic missions abroad can provide necessary short language courses and provide vital cultural information during the granting of study permits. If necessary, the corresponding diplomatic representatives and their resident citizens in Bangladesh can cooperate to integrate the students into the community. They can also guide them to blend in with the culture. The dental college authorities also should arrange regular parents’ meetings (on-site and virtual) and open dialogues between the students, alumni, and their country representatives in Bangladesh. It is also imperative that educational institutes have clubs and events that allow for the mixing of IUSs from other institutes/universities. Also, routine psychological counseling, food and nutrition guidelines, and academically encouraging seminars should be arranged for the IUDSs. As dental education requires interaction with patients, special classes could be set for the IUDSs to learn the basic language skills and cultural knowledge of the local community.

## Figures and Tables

**Table 1 ijerph-18-05802-t001:** Association between background characteristics and depression among the international students studying dentistry in Bangladesh.

Variables	Total *n* (%)	No Depression *n* (%)	Depression *n* (%)
**Total**	**200 (100)**	**97 (48.5)**	**103 (51.5)**
Age Group			
18–24 years	159 (79.5)	80 (50.3)	79 (49.6)
More than 24 years	41 (20.5)	17 (41.5)	24 (58.5)
Gender			
Female	157 (78.5)	75 (47.8)	82 (52.2)
Male	43 (21.5)	22 (51.2)	21 (48.8)
Religion			
Muslim	70 (35)	36 (51.4)	34 (48.6)
Others	130 (65)	61 (46.9)	69 (53.1)
Marital Status			
Married	17 (8.5)	8 (47.1)	9 (52.9)
Single	183 (91.5)	89 (48.6)	94 (51.4)
Year of Study			
1st year	24 (12)	14 (58.3)	10 (41.7)
2nd year	26 (13)	14 (53.8)	12 (46.2)
3rd year	36 (18)	14 (38.9)	22 (61.1)
4th year	84 (42)	41 (48.8)	43 (51.2)
Intern	30 (15)	14 (46.7)	16 (53.3)
Living Arrangement in Bangladesh			
Living alone	16 (8)	4 (25.0)	12 (75.0)
Living with friends	170 (85)	90 (52.9)	80 (47.1)
Living with others	14 (7)	3 (21.4)	11 (78.6)
Communication in Bengali (Bangla) Language			
Good	135 (67.5)	71 (52.6)	64 (47.4)
Not good	65 (32.5)	26 (40.0)	39 (60.0)
Problem Regarding Studying			
No	81 (40.5)	53(65.4)	28 (34.6)
Yes	119 (59.5)	44 (37.0)	75 (63.0)
Teacher and Student’s Academic Interaction			
Fair	126 (63)	61 (48.4)	65 (51.6)
Full	35 (17.5)	24 (68.6)	11 (31.4)
Poor	39 (19.5)	12 (30.8)	27 (69.2)

**Table 2 ijerph-18-05802-t002:** Challenges faced by the IUDSs and the association with depression—Chi-squared tests.

Challenges Faced	No Depression *n* (%)	Depression *n* (%)	*p*-Value	Cramer’s V
Problem adapting local Bangladeshi food	34 (29.8)	80 (70.2)	0.11	0.15
Homesickness	17 (19.5)	70 (80.5)	0.00	0.33
Health problems	26 (23.2)	131 (65.5)	0.00	0.35
Financial difficulty	17 (22.7)	58 (77.3)	0.00	0.27
Accommodation problems	12 (20.3)	47 (79.7)	0.00	0.18
Transport problems	41 (34.5)	78 (65.5)	0.98	0.12

**Table 3 ijerph-18-05802-t003:** Association between challenges faced by the IUDSs and depression― multivariable logistic regression analysis.

Challenges Faced	OR (95% CI)	*p*-Value
Problems adapting to local Bangladeshi food		
No	1	0.083
Yes	1.07 (0.55–2.12)	
Accommodation problems		
No	1	0.69
Yes	1.00 (0.54–2.52)	
Homesickness		
No	1	0.00
Yes	2.55 (1.30–4.99)	
Health problems		
No	1	0.00
Yes	3.58 (1.89–6.82)	
Financial difficulty		
No	1	0.01
Yes	2.38 (1.20–4.70)	
Transport problems		
No	1	0.84
Yes	0.11 (0.54–2.08)	

Nagelkerke R Square = 0.307; Hosmer and Lemeshow Test (*p* = 0.783).

## Data Availability

The data presented in this study are available on reasonable request from the corresponding author.

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
