# Peer review of "Depression among the Non-Native International Undergraduate Students Studying Dentistry in Bangladesh"

_ijerph, 2021, doi:10.3390/ijerph18115802_

Round 1

Reviewer 1 Report

see attached file, please.

Author Response

ABSTRACT

Indicate type of sampling carried out. The sample size (N = 200) is very small bearing in mind the age range. Indicate the percentage of men or women recruited for this study, as well as the age range, mean and SD. What does financial problems mean? Provide operational definition of this variable.

Thanks for your comments. We have added the information in the abstract. See page 1 line 28-29 and 32. Highlighted in yellow.

INTRODUCTION

The review of previous studies should be better founded. Thus, authors should focus on citing and describing studies previously published in or in Bangladesh. In this sense, this section should end by indicating the general objective of this ms., The specific objectives and the hypotheses related to each specific objective, which should be supported by the previous empirical evidence found in this type of population. Furthermore, these hypotheses should be accepted or rejected in the Discussion section, providing robust reasons in each case.

Thanks for your comments. We have done substantial changes to the introduction. Page 2, line 51, 52, 66-74, 97-99. Highlighted in yellow.

METHODS

Describe exhaustively what the type of sampling conducted in this ms consists of. (i.e. snowballing sampling).

Done. See the manuscript  page 3 line 117. Highlighted in yellow.

RESULTS

The logistic regression analysis presented is not correctly described and interpreted. The results presented are not accompanied by the corresponding effect sizes. This is a serious error because the authors can reach conclusions that are not rigorous and accurate from a point of view in the Discussion section.

Thanks for your comments. We have added a description of logistic regression analysis in the methods section (page ..3 line 140- 147.) . We have also updated our interpretation in the results section (page .4 and 5..) . Changes are highlighted in yellow.

We have reported odds ratio with 95% confidence interval along with p value. Please see Table 3 and corresponding interpretation (page ...). OR are the evidence of the strength of association between depression and selected explanatory variables.      

Reviewer 2 Report

Dear authors.

Thank you very much for your valuable work.

I would like to offer some suggestions about your paper.

First of all, introduction section doesn't provide enough information. I think that you should add some information about depression or educational stress models. 

Besides, I think that you should provide information about your hypotheses and its theoretical justification.

In discussion section, I think that you should add some of your study limitations and suggestions for future research. 

Methods and results sections are fine.

I hope you find my suggestions useful.

Good luck in publishing.

Author Response

First of all, introduction section doesn't provide enough information. I think that you should add some information about depression or educational stress models.

Thanks for your comments. We have added information depression and educational stress models in the introduction highlighted in yellow. See Page 2, line 51, 52, 66-74, 97-99, 

Besides, I think that you should provide information about your hypotheses and its theoretical justification.

Done. See the manuscript page 3 line 103-105

In discussion section, I think that you should add some of your study limitations and suggestions for future research.

Added. Please see the manuscript

Round 2

Reviewer 1 Report

The results presented in Tables 2 and 3 are not completed with their corresponding effect sizes. In this line, Discussion section does not correctly interpreted the findings of this ms. because the authors do not take effect sizes into account. It is serious mistake.

Author Response

The results presented in Tables 2 and 3 are not completed with their corresponding effect sizes. In this line, Discussion section does not correctly interpreted the findings of this ms. because the authors do not take effect sizes into account. It is serious mistake.

Thanks for your comments.Odds ratio is a measure of effect size. Although there are other effect size measures, reporting OR is the most common effect size measure reported in public health research articles. OR is widely understood by public health practitioners, researchers and academicians. Therefore, we believe we did not commit any serious error by not reporting other effect size measures. To support our response, we request the reviewer and the editor to randomly select articles published in ISI indexed public health journals, including the ijerph, and check reporting practices in this regard. We are confident that you will find that more than 90% articles only reported OR with 95% CI.In revision, we included model fit information for our multivariable logistic regression analysis. Please see Table 3.

Please read the attached article.It mentioned: The odds ratio (OR) is probably the most widely used index of effect size in epidemiological studies. 

https://www.tandfonline.com/doi/full/10.1080/03610911003650383

Regarding extensive editing of English, language-We have re-read the manuscript carefully for any errors in grammar or spelling. Some of the expressions represent the style of scientific writing, which may differ from the style of the reviewers. Three of the authors have been working on academic research for several years at the reputed universities in the USA and the UK.

This manuscript is a resubmission of an earlier submission. The following is a list of the peer review reports and author responses from that submission.